# Winnie-*APC^Min/+^* Mice: A Spontaneous Model of Colitis-Associated Colorectal Cancer Combining Genetics and Inflammation

**DOI:** 10.3390/ijms21082972

**Published:** 2020-04-23

**Authors:** Stefania De Santis, Giulio Verna, Grazia Serino, Raffaele Armentano, Elisabetta Cavalcanti, Marina Liso, Manuela Dicarlo, Sergio Coletta, Mauro Mastronardi, Antonio Lippolis, Angela Tafaro, Angelo Santino, Aldo Pinto, Pietro Campiglia, Alex Y. Huang, Fabio Cominelli, Theresa T. Pizarro, Marcello Chieppa

**Affiliations:** 1Department of Pharmacy-Drug Science, University of Bari Aldo Moro, 70126 Bari, Italy; 2Department of Pharmacy, University of Salerno, 84084 Fisciano (SA), Italy; pintoal@unisa.it (A.P.); pcampiglia@unisa.it (P.C.); 3National Institute of Gastroenterology “S. de Bellis”, Research Hospital, 70013 Castellana Grotte (BA), Italy; giu.verna@gmail.com (G.V.); grazia.serino@irccsdebellis.it (G.S.); raffaele.armentano@irccsdebellis.it (R.A.); elisabetta.cavalcanti@irccsdebellis.it (E.C.); marinaliso@libero.it (M.L.); manueladicarlo@alice.it (M.D.); sergiofalaut@hotmail.it (S.C.); mauro.mastronardi@irccsdebellis.it (M.M.); antonio.lippolis@irccsdebellis.it (A.L.); angela.tafaro@irccsdebellis.it (A.T.); 4Institute of Sciences of Food Production C.N.R., Unit of Lecce, 73100 Lecce, Italy; angelo.santino@ispa.cnr.it; 5Division of Pediatric Hematology-Oncology, Department of Pediatrics, Angie Fowler AYA Cancer Institute, Cleveland, OH 44106, USA; ayh3@case.edu; 6Digestive Health Research Institute, Case Western Reserve University School of Medicine, Cleveland, OH 44106, USA; fabio.cominelli@case.edu (F.C.); theresa.pizarro@case.edu (T.T.P.); 7Department of Pathology, Case Western Reserve University School of Medicine, Cleveland, OH 44106, USA; 8European Biomedical Research Institute of Salerno (EBRIS), 84125 Salerno, Italy

**Keywords:** colorectal cancer, murine model, inflammation, Aberrant Crypt Foci

## Abstract

(1) Background: Colorectal cancer (CRC) is among the best examples of the relationship between inflammation and increased cancer risk. (2) Methods: To examine the effects of spontaneous low-grade chronic inflammation on the pathogenesis of CRC, we developed a new murine model of colitis-associated cancer (CAC) by crossing Mucin 2 mutated mice (Winnie) with *Apc^Min/+^* mice. (3) Results: The resulting Winnie-*Apc^Min/+^* model combines an inflammatory background with a genetic predisposition to small intestinal polyposis. Winnie-*Apc^Min/+^* mice show an early occurrence of inflammatory signs and dysplastic lesions in the distal colon with a specific molecular signature. (4) Conclusion: The Winnie-*Apc^Min/+^* model is a perfect model to demonstrate that chronic inflammation represents a crucial risk factor for the onset and progression of tumoral lesions in individuals genetically predisposed to CRC.

## 1. Introduction

Colorectal cancer (CRC) is the third most commonly diagnosed cancer and the fourth leading cause of cancer-related mortality worldwide, becoming a global public health problem with social and economic implications [1]. The development of CRC is influenced by several factors, such as genetics, environmental factors and lifestyle, including diet. The exposure to toxic signals or poor habits, such as following the Western diet, could induce an increase in the inflammatory state, thereby predisposing an individual to CRC, as well as to other types of cancer. In fact, the relationship between cancer and chronic inflammation was first described by Virchow a long time ago (reviewed in [2,3]) and later supported by the introduction of non-steroidal anti-inflammatory drugs for the treatment of chronic inflammatory disorders. These treatments were able to protect patients from an increased risk of developing various forms of cancer [4]. Specifically, in the gastrointestinal tract, before the introduction of effective therapeutic options for controlling chronic inflammation, inflammatory bowel disease (IBD) was associated with an increased risk of developing CRC [5]. For this reason, CRC could represent an ideal field of investigation to study the link between inflammation and cancer. In fact, CRC was first recognized as a complication of both Crohn’s disease [6] and ulcerative colitis (UC) by Rosenberg in 1925 [7]. Although CRC in UC patients only accounts for 1% of all cases reported in the general population, it also represents one-sixth of all deaths in UC patients [8]. In more recent years, numerous epidemiological studies have confirmed an increased risk of CRC for UC patients in the first decade after UC diagnosis [9]. Although the risk in adults continues to be a matter of debate, the general consensus from all studies is that the risk of CRC is higher in patients who are diagnosed with UC during childhood, adolescence or young adulthood, especially in male patients [10,11].

To dissect the complex relationship between inflammation and increased CRC risk, we took advantage of two established mouse models: *Apc^Min/+^* mice, a commonly used model of genetically predisposed small intestinal polyposis [12], and Winnie mice, a low-grade, spontaneous and progressive model of UC [13]. Winnie mice carry a single missense mutation (G9492A, GenBank accession no. AJ511872) that results in the substitution of cysteine with tyrosine in the D3 domain in the N terminus of the *Mucin 2* (*Muc2)* gene. This mutation induces aberrant Muc2 biosynthesis, a thinner mucus barrier with increased intestinal permeability, and enhanced local production of inflammatory cytokines in the distal colon. Winnie mice differ from Muc2-deficient mice, which display an aberrant intestinal crypt morphology and altered cell maturation and migration. Another important difference is the ability of Muc2-deficient mice to frequently develop adenomas in the small intestine, which can progress to invasive adenocarcinoma, as well as to rectal tumors [14]. 

In the present study, we developed the Winnie-*Apc^Min/+^* mouse strain, a new model of colitis-associated cancer, and carefully characterized the development of intestinal neoplastic lesions during the first weeks of life, along with a full characterization of the molecular profile. Our results demonstrate that even low-grade, yet chronic, inflammation could promote the development of CRC in genetically predisposed mice, supporting its progression over time.

## 2. Results

### 2.1. Creation and Characterization of the New Spontaneous Model of Colitis-Associated Colorectal Cancer: The Winnie-APC^Min/+^ Model 

With the intent to create the Winnie-*Apc^Min/+^* model, we performed a breeding strategy using mice carrying double heterozygote mutations for males (on the *Apc* and *Muc2* genes in *Apc^Min/+^* and Winnie mice, respectively) and a single heterozygote mutation (on the *Muc2* gene in Winnie mice) for females (Figure 1A).

This strategy generated a model of colitis-associated CRC (CAC) (the Winnie-*Apc^Min/+^* model), together with all the parental strains (WT, Winnie and *Apc^Min/+^* mice) from the same breeding pairs for which survival (Figure 1B) and inflammatory features (Figure 1C–F) were evaluated. Comparing breeders used to obtain Winnie and Winnie-*Apc^Min/+^* mice, we demonstrated that the *Apc^Min/+^* mutation induced a dramatic reduction in the survival rate (28.6% vs. 10.8%, respectively; Figure 1B) and an overall decrease in the fertility rate (Appendix A). In addition, 5-week-old Winnie-*Apc^Min/+^* mice, similar to native Winnie mice, were characterized by a reduction in body weight (Figure 1C) and colon length (Figure 1D,E) when compared to WT and *APC^Min^*^/+^ control mice. These data were emphasized when the colon length was related to mouse weight (Figure 1F). 

### 2.2. The Novel Winnie-Apc^Min/+ -^Mouse Strain Develops Colitis-Associated CRC Uniquely in the Colon with Early Appearance of Aggressive Dysplastic Lesions Compared to Parental Controls 

Histological analysis of 5-week-old Winnie-*Apc^Min/ +^* mice revealed the presence of epithelial erosion similar to that observed in Winnie mice, as well as the presence of dysplastic aberrant crypt foci (ACFs) along the entire colon length, with a gradual increase in incidence, multiplicity and tumor grade moving from the proximal to the distal colon (Figure 2 and Table 1). 

As expected, dysplastic ACFs were absent in all age-matched parental control strains, with the exception of *Apc^Min/+^* mice that showed occasional ACFs, although with lower multiplicity compared to Winnie-*Apc^Min/+^* mice (30% vs. 78% incidence and multiplicity of 0.3 ± 0.15 vs. 2.2 ± 0.62, respectively) (Table 1). Dysplastic ACFs were carefully classified according to incidence and multiplicity into four groups, based on the number of crypts and tumor grading: unicryptic lesions; microadenoma >1 ≤ 5 crypts low-grade (LG); microadenoma >5 crypts low-grade (LG) and microadenoma >5 crypts high-grade (HG) (Table 1). All tumor lesions showed a tubular pattern. *Apc^Min/+^* and Winnie-*Apc^Min/+^* mice also showed low incidence and multiplicity of non-dysplastic ACFs within the proximal and medial portions of the colon, respectively (Table 1). 

Periodic acid Schiff (PAS) staining confirmed an overall decrease in mucin expression in the distal colon of Winnie-*Apc^Min/+^* mice compared to WT and *Apc^Min/+^* mice, in line with the characteristics described for the Winnie parental line [15]. Moreover, PAS staining in the dysplastic area replicated the characteristics described for human CRC [16] (Figure 3). 

### 2.3. Gene Expression Profile from 5-Week-Old Winnie-Apc^Min/+^ Mice Shows Reduced Apoptosis and Increased Cellular Proliferation and Cytoskeletal Remodeling Patterns

RNA from the distal colon of 5-week-old Winnie-*Apc^Min/+^* mice and age-matched parental strains was analyzed by qPCR to measure the expression of 89 selected genes specific for CRC. By comparing Winnie-*Apc^Min/+^* with Winnie and *Apc^Min/+^* mouse strains, we identified genes uniquely modulated in CAC, thereby demonstrating the contribution of Winnie and *Apc^Min/+^* mutations in this process. Molecular analysis of Winnie-*Apc^Min/+^* mice showed increased cellular proliferation and decreased apoptosis, along with cytoskeletal remodeling (Figure 4). 

Furthermore, we identified a more pronounced modulation of genes involved in inflammatory processes as compared to those contributing to CRC development at early time points. In fact, a Venn diagram showed that 26 vs. 8 genes were differentially regulated in Winnie and *Apc^Min/+^*, respectively, compared to WT mice (Appendix A). Specifically, among the 26 significantly modulated genes obtained by comparing Winnie vs. WT mice, 16 were shared with Winnie-*Apc^Min/+^* mice (10 up- and 6 down-modulated); in contrast, only one gene from the comparison between *Apc^Min/+^* and WT showed a similar modulation to Winnie-*Apc^Min/+^* mice (down, i.e., *Ifng*). Interestingly, among the 16 genes in common with Winnie-*Apc^Min/+^* mice, we found 7/10 up- (*Plau, Plaur, Ptgs2, Rela, Sfn, Myc, Ccnd2*) and 3/6 down-modulated (*Ar, Acta2, Kit*) genes that were also significantly regulated in the comparison of Winnie-*Apc^Min/+^* to *Apc^Min/+^* mice that underlined the role of Winnie mutations (Figure 4B). Conversely, common modulation for *Ifng* was not found when comparing Winnie-*Apc^Min/+^* to Winnie mice (Figure 4A). In addition, when comparing Winnie and *Apc^Min/+^* mice to WT, only two genes (*Cxcl10* and *Egr1*) were in common in these two murine lines, although they showed opposite regulation (Appendix A). Moreover, we demonstrated that only three genes (*Il1**β*, *Cd44* and *Mmp2*) were in common in all three analyses performed starting from the same data set (Winnie-*Apc^Min/+^* vs. Winnie, *Apc^Min/+^* and WT mice) (Figure 4, horizontal lines and Appendix A). In fact, these 3 genes were also among the 10 genes specifically up-regulated in Winnie-*Apc^Min/+^* vs. WT mice (Appendix A), indicating that the up-regulation of *Il1b*, *Cd44* and *Mmp2* was specific for Winnie-*Apc^Min/+^* mice. 

### 2.4. Winnie-Apc^Min/+^ Model Confirms the Multi-Step Nature of CRC Increasing the Multiplicity and Tumor Grading of Dysplastic ACFs in Older 8-Week-Old Mice 

In line with the multi-step nature of CRC, we conducted histological characterization at a later time point to confirm the expected progression in the number of ACFs of 8-week-old Winnie-*Apc^Min/+^* mice (Figure 5A,B). 

Together with the increased rate of progression, the type of ACFs and their tumor grade were also altered over time. Specifically, small neoplastic lesions (unicryptic lesions and microadenoma >1 ≤ 5 crypts LG) appeared less frequently, while a marked increase in microadenoma >5 crypts LG was observed when comparing 5- and 8-week-old Winnie-*Apc^Min/+^* mice (Table 1 and Table 2 and Figure 5C). Thus, the histological analysis of the Winnie-*Apc^Min/+^* colon at 8 weeks showed a marked increase in the multiplicity, dimension and grading of dysplastic ACFs (Figure 5). 

## 3. Discussion 

Colorectal cancer is among the best examples of the relationship between inflammation and increased cancer risk. Despite the use of therapies to better control inflammation and routine endoscopic screening for adult IBD patients to efficiently reduce CRC risks, some epidemiological studies still reveal an association between IBD and CRC [17]. Of note, the axis between intestinal inflammation and cancer development is especially true for patients diagnosed with IBD during the pediatric years, particularly for UC patients [9,10,18].

In the present study, we developed a unique mouse model of UC-induced CRC, the Winnie-*Apc^Min/+^* model, that recapitulates the pathology and clinical progression in humans. This model combines the strengths and reduces the limitations of the best-known models of CRC, which are the azoxymethane/dextran sodium sulfate (AOM/DSS) model and the *Apc^Min/+^* model [19]. In fact, even if AOM-induced colorectal cancer could reproduce the human pathology, it requires several weeks and repeated cycles of acute inflammatory insults via DSS administration to induce the development of tumors in the distal colon [20]. Regardless, the use of two chemicals in the AOM/DSS model does not represent human CRC with regard to etiology. On the other hand, the *Apc^Min/+^* model carries a germline mutation in the *APC* gene that results in familial adenomatous polyposis (FAP), an autosomal dominant disorder characterized by the presence of multiple adenomatous polyps with a high likelihood of developing colorectal carcinomas [12]. However, the *Apc^Min/+^* model uniquely develops polyps in the small intestine of affected mice, not in the colon. In contrast with previously generated murine models that combine *Apc^Min/+^* with other mutations that lead to adenocarcinoma development [21], the Winnie-*Apc^Min/+^* model relies on mild but chronic inflammation to promote cancer development in genetically susceptible subjects. Indeed, Winnie mice are characterized by mild but progressive intestinal inflammation with well-defined inflammatory steps that are often detectable in the intestinal tract before the appearance of morpho-pathological signs of ulcerative colitis [22,23,24,25,26].

Specifically, the presence of ACFs, along with a reduction in mucin expression in the dysplastic areas in Winnie-*Apc^Min/+^* mice, strengthens the translational value of our model by demonstrating its ability to resemble the human pathology. In fact, ACFs in the colon could be considered to be precursors of adenoma and carcinoma in patients with CRC [16]. Thus, this model recapitulates the location, progression and histological features of human CAC in a very short amount of time without the need for any immunological or chemical manipulation. 

Our model also recapitulates a colorectal cancerogenesis process from a molecular point of view. In fact, the molecular analysis of selected genes for CRC development showed a specific modulation in terms of cellular proliferation, apoptosis and cytoskeletal pathways in Winnie-*Apc^Min/+^* mice. The positive effects on cellular proliferation were induced by the down-regulation of oncogenes and the up-modulation of tumor suppressor genes. Two exceptions should be noted. The first one is in regards to the *Cdkn2a* gene, which encodes for the CDK4/6 inhibitor p16ink4a. *Cdkn2a* was found to be reduced in many tumors, but in others, its overexpression was associated with aggressive subtypes of disease, and it could be useful for disease prognosis and therapeutic response in certain clinical settings [27]. Another exception has to be made for the oncogene *Kit*, the reduction of which was found in association with CRC development in a large set of human colorectal cancer tissue samples [28]. Moreover, confirming the macroscopic features and the histological data, Winnie-*Apc^Min/+^* mice shared a more similar molecular profile with Winnie than with *Apc^Min/+^* mice. In fact, more than half (16/26) and only one (1/8) of the significantly modulated genes from Winnie and *Apc^Min/+^* vs. WT mice analysis, respectively, were also modulated in the comparison between Winnie-*Apc^Min/+^* and WT mice. This data could also be explained considering a more limited number of significantly regulated genes in *Apc^Min/+^* vs. WT mice, which is in line with a very preliminary molecular analysis of this murine line that needs more time to develop its intrinsic tumorigenic features.

The dissection of the molecular pathways in the Winnie-*Apc^Min/+^* model paves the way for the identification of new biomarkers for early CRC diagnosis in IBD patients, and this could be particularly useful for the design of alternative therapies. In fact, the lack of response to biological agents is one of the major drawbacks of this new frontier of drugs, with an enormous financial cost for the health system. 

Overall, considering the ability of the Winnie-*Apc^Min/+^* model to combine genetics and inflammation, it could be considered a unique tool to develop new pharmaceutical- and nutraceutical-based strategies to prevent and/or reduce CRC in genetically predisposed individuals. 

## 4. Material and Methods

### 4.1. Mice

The animal studies were conducted in accordance with national and international guidelines and were approved by the authors’ institutional review board (Organism For Animal Wellbeing, OPBA). All animal experiments were carried out in accordance with Directive 86/609 EEC, enforced by Italian D.L. n. 116/1992, and approved by the Committee on the Ethics of Animal Experiments of Ministero della Salute - Direzione Generale Sanità Animale (Prot. 768/2015-PR 27 July 2015) and the official RBM veterinarian. Any animals found in severe clinical condition were sacrificed in order to avoid undue suffering.

The new murine transgenic line Winnie-*Apc^Min/+^* was created by breeding Winnie mice with *Apc^Min/+^* mice on a C57BL/6J background. WT and *Apc^Min/+^* murine lines were purchased from Jackson Laboratories (C57BL/6J, Stock No. 000664, C57BL/6J-ApcMin/J, Stock No. 002020, respectively) (Bar Harbor, ME, USA). Winnie mice were obtained from the University of Tasmania (Dr R Eri’s laboratory). 

Mice were sacrificed at 5 or 8 weeks of age, and their colons were removed to evaluate the presence of neoplasia. Colon lengths and weights were measured as indicators of colonic inflammation. 

### 4.2. Histology

Tissue sections from the large intestine were fixed in 10% buffered formalin, dehydrated and paraffin-embedded. Three-micrometer-thick sections from the proximal, medial and distal colon were stained using a hematoxylin and eosin standard protocol. Colonic tissue sections were evaluated for neoplasia. PAS staining on distal colon sections was performed to identify mucins. Observations and imaging were performed with a Nikon Eclipse Ti2. 

### 4.3. RNA Extraction from Colon Tissue and qPCR Analysis

Total RNA was isolated from the distal part of the large intestine of mice. The RNA was extracted using TRIzol^®^ (Thermo Fisher Scientific, Waltham, MA, USA) according to the manufacturer’s instructions. Total RNA (1 µg) was reverse transcribed using an iScript cDNA Synthesis kit (Biorad, Hercules, CA, USA) with random primers for cDNA synthesis. Gene expression analysis was performed using Colonic neoplasms Tier 1 M96 (Cat.#100–36551, Biorad, CA, USA). Real-time PCR was performed on the CFX96 System (Biorad, Hercules, CA, USA), and the expression of all target genes was calculated relative to *Gapdh* expression using the 2 ^−ΔΔCt^ method. 

### 4.4. Statistical Analysis

Statistical evaluation was performed using the GraphPad Prism statistical software release 6.01. All data are expressed as the mean ± SD or SEM. All results were obtained from three consecutive and independent experiments. Unless specifically described, all data were analyzed and compared by paired, two-tailed Student’s *t*-tests. Results were considered statistically significant at *p* < 0.05. 

## Figures and Tables

**Figure 1 ijms-21-02972-f001:**
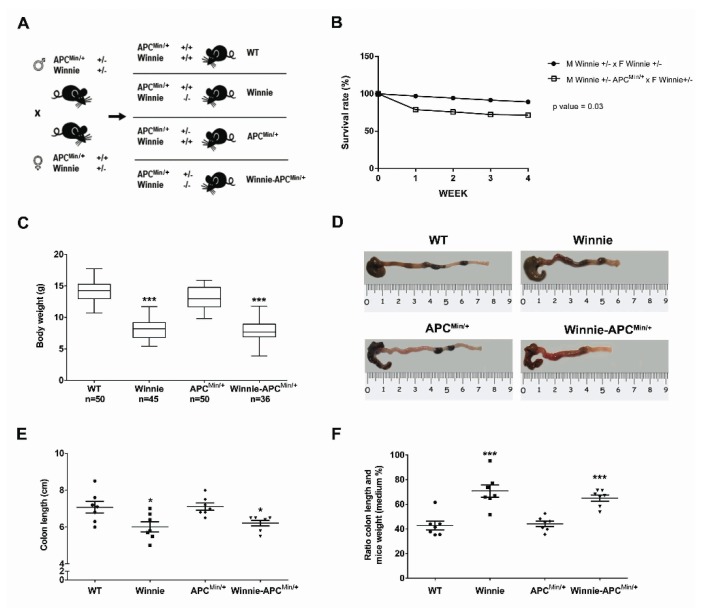
Creation and characterization of the Winnie-*APC^Min/+^* murine model. (**A**) Breeding strategy used to obtain Winnie-*APC^Min/+^* mice and the control littermates. (**B**) Impact of *APC^Min/+^* mutation on survival rate in the breeding scheme of Winnie and Winnie-*APC^Min/+^* mice. (**C**) Body weight analysis of 4-week-old Winnie-*APC^Min/+^* mice and their relative controls. (**D**–**F**) Representative images of colon (**D**) and data analysis of colon length, and (**E**) colon length adjusted to body weight (**F**) for Winnie-*APC^Min/+^* mice and control lines sacrificed at 5 weeks. (**E**,**F**): *n* = 7 animals/group. * *p* < 0.05, *** *p* < 0.001 compared to WT mice.

**Figure 2 ijms-21-02972-f002:**
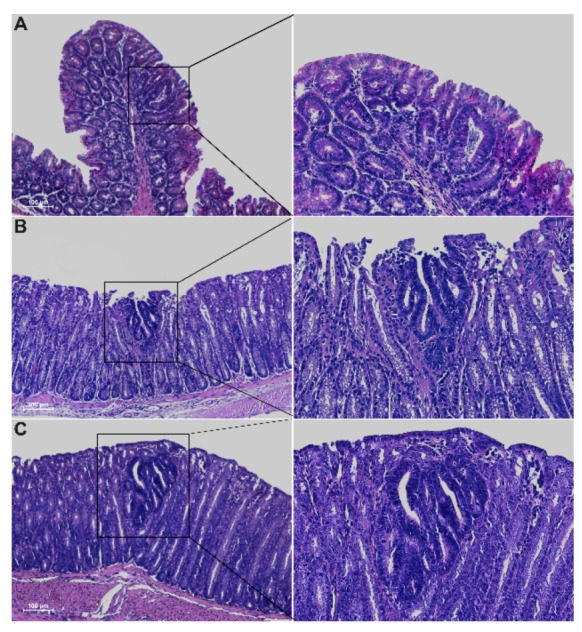
Hematoxylin and eosin staining on 3μm colon sections from proximal (**A**), medial (**B**) and distal (**C**) tracts of 5-week-old Winnie-*APC^Min/+^* mice. Images were captured at 10× (left) and 20× magnifications (right).

**Figure 3 ijms-21-02972-f003:**
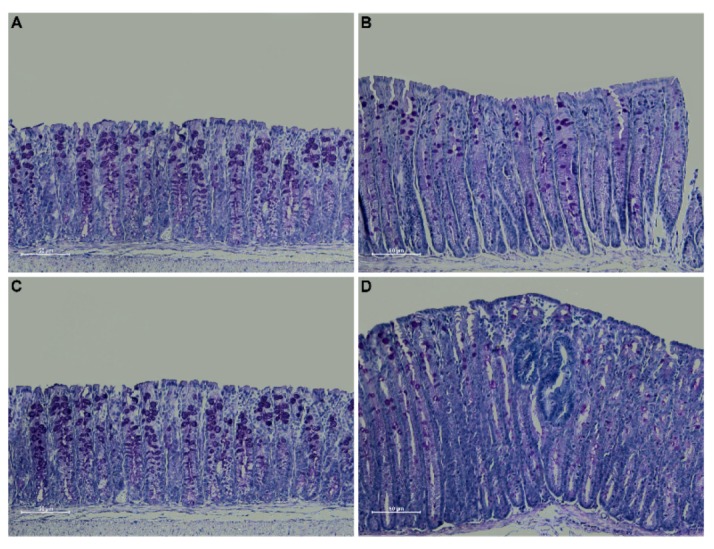
PAS staining on 3 μm sections from distal colon of 5-week-old WT (**A**), Winnie (**B**), *APC^Min/+^* (**C**) and Winnie-*APC^Min/+^* (**D**) mice. Images were captured at 20× magnification.

**Figure 4 ijms-21-02972-f004:**
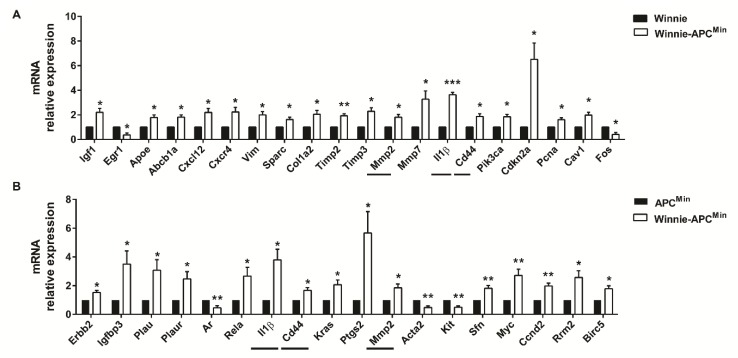
Molecular pathways activated in 5-week-old Winnie-*APC^Min/+^* mice. The distal colon of Winnie-*APC^Min/+^* mice (white bars) was analyzed by qPCR relative to Winnie (**A**) and *APC^Min/+^* (**B**) mice (black bars) (*n* = 3–4 animals/group). Horizontal lines indicate common genes between the two comparisons. Histograms represent the mean ± SEM. * *p* < 0.05, ** *p* < 0.01, *** *p* < 0.001.

**Figure 5 ijms-21-02972-f005:**
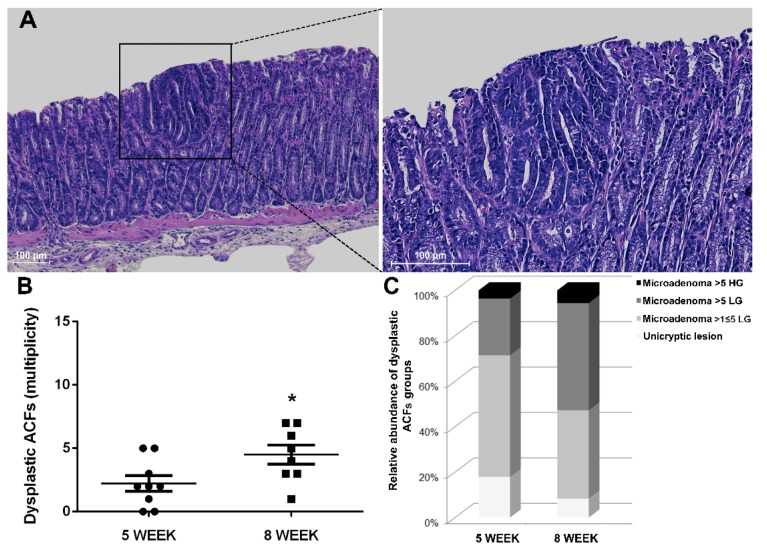
Histological analysis of 8-week-old Winnie-*APC^Min/+^* mice. (**A**) Hematoxylin and eosin staining on 3μm distal colon sections. Images were captured at 10× (left) and 20× magnifications (right). (**B**) Dot plot shows CRC progression from 5 to 8 weeks for Winnie-*APC^Min/+^* mice as mean ± SEM of dysplastic ACFs. (**C**) Histograms show dysplastic ACFs composition in terms of dimension and grading for Winnie-*APC^Min/+^* mice at 5 and 8 weeks. 5-week: *n* = 8–10 animals/group; 8-week: *n* = 8 animals/group. **p* < 0.05.

**Table 1 ijms-21-02972-t001:** Histological analysis of 5-week-old Winnie-*APC^Min/+^* mice and control littermates. Incidence and multiplicity ± SEM of proximal, medial and distal colon were calculated for non-dysplastic and dysplastic aberrant crypt foci (ACFs). The score for dysplastic ACFs was calculated relative to all groups and to each group of neoplastic lesions classified according to dimension and grading of unicryptic lesions; microadenoma >1 ≤ 5 crypts low-grade (LG); microadenoma >5 crypts low-grade (LG) and microadenoma >5 crypts high-grade (HG).

5-WEEK	Non-Dysplastic ACFs		Dysplastic ACFs									
			All Groups		Unicryptic Lesion		Microadenoma >1 ≤ 5-LG		Microadenoma >5-LG		Microadenoma >5-HG	
Genotype (nr. mice)	Incidence (%)	Multiplicity (mean ± SEM)	Incidence (%)	Multiplicity (mean ± SEM)	Incidence (%)	Multiplicity (mean ± SEM)	Incidence (%)	Multiplicity (mean ± SEM)	Incidence (%)	Multiplicity (mean ± SEM)	Incidence (%)	Multiplicity (mean ± SEM)
**Proximal Colon**												
WT (8)	0	0	0	0	0	0	0	0	0	0	0	0
Winnie (8)	0	0	0	0	0	0	0	0	0	0	0	0
*APC^Min/+^* (10)	20	0.2 ± 0.13	0	0	0	0	0	0	0	0	0	0
Winnie-*APC^Min/+^* (9)	0	0	11.1	0.1 ± 0.1	100	0.1 ± 0.1	0	0	0	0	0	0
**Medial Colon**												
WT (8)	0	0	0	0	0	0	0	0	0	0	0	0
Winnie (8)	0	0	0	0	0	0	0	0	0	0	0	0
*APC^Min/+^* (10)	0	0	0	0	0	0	0	0	0	0	0	0
Winnie-*APC^Min/+^* (9)	10	0.2 ± 0.2	66.7	1.1 ± 0.4	10	0.1 ± 0.1	70	0.78 ± 0.28	20	0.22 ± 0.15	0	0
**Distal Colon**												
WT (8)	0	0	0	0	0	0	0	0	0	0	0	0
Winnie (8)	0	0	0	0	0	0	0	0	0	0	0	0
*APC^Min/+^* (10)	0	0	30	0.3 ± 0.15	33.3	0.1 ± 0.1	66.7	0.2 ± 0.13	0	0	0	0
Winnie-*APC^Min/+^* (9)	0	0	77.8	2.2 ± 0.62	20	0.44 ± 0.24	50	1.1 ± 0.6	25	0.6 ± 0.24	5	0.1 ± 0.1

**Table 2 ijms-21-02972-t002:** Histological analysis of 8-week-old Winnie-*APC^Min/+^* mice and control littermates. The incidence and multiplicity ± SEM of proximal, medial and distal colon were calculated for non-dysplastic and dysplastic ACFs. The score for dysplastic ACFs was calculated relative to all groups and to each group of neoplastic lesions classified according to dimension and grading of unicryptic lesions; microadenoma >1 ≤ 5 crypts low-grade (LG); microadenoma >5 crypts low-grade (LG) and microadenoma >5 crypts high grade (HG).

8-WEEK	Non-Dysplastic ACFs		Dysplastic ACFs									
			All Groups		Unicryptic Lesion		Microadenoma >1 ≤ 5-LG		Microadenoma >5-LG		Microadenoma >5-HG	
Genotype (nr. mice)	Incidence (%)	Multiplicity (mean ± SEM)	Incidence (%)	Multiplicity (mean ± SEM)	Incidence (%)	Multiplicity (mean ± SEM)	Incidence (%)	Multiplicity (mean ± SEM)	Incidence (%)	Multiplicity (mean ± SEM)	Incidence (%)	Multiplicity (mean ± SEM)
**Proximal Colon**												
WT (8)	0	0	0	0	0	0	0	0	0	0	0	0
Winnie (8)	0	0	0	0	0	0	0	0	0	0	0	0
*APC^Min/+^* (8)	0	0	0	0	0	0	0	0	0	0	0	0
Winnie-APC*^Min/+^* (8)	0	0	0	0	0	0	0	0	0	0	0	0
**Medial Colon**												
WT (8)	0	0	0	0	0	0	0	0	0	0	0	0
Winnie (8)	0	0	0	0	0	0	0	0	0	0	0	0
*APC^Min/+^* (8)	0	0	0	0	0	0	0	0	0	0	0	0
Winnie-APC*^Min/+^* (8)	0	0	70	1.4 ± 0.47	13.3	0.2 ± 0.13	40	0.55 ± 0.26	26.7	0.4 ± 0.2	20	0.3 ± 0.2
**Distal Colon**												
WT (8)	0	0	0	0	0	0	0	0	0	0	0	0
Winnie (8)	0	0	0	0	0	0	0	0	0	0	0	0
*APC^Min/+^* (8)	0	0	37.5	0.38 ± 0.18	0	0	33.4	0.13 ± 0.13	33.3	0.13 ± 0.13	33.3	0.13 ± 0.13
Winnie-*APC^Min/+^* (8)	0	0	100	4.8 ± 0.77	10.4	0.5 ± 0.2	43.8	2.1 ± 0.4	41.7	2 ± 0.6	4.2	0.2 ± 0.13

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
