# Peer review of "Winnie-APCMin/+ Mice: A Spontaneous Model of Colitis-Associated Colorectal Cancer Combining Genetics and Inflammation"

_ijms, 2020, doi:10.3390/ijms21082972_

Round 1

Reviewer 1 Report

This manuscript describes a new mouse model for inflammation-associated colon cancer by combining a mutation in the gene coding for the mucin Muc2 with the Min mutation in the Apc gene. The data show that the presence of the mutant Muc2 in ApcMin mice increases the incidence of polyps in the distal colon in mice as young as 5 weeks of age.

The studies are certainly interesting and important, but provide only minimal information on the biochemical and cell-biological features of the model.

Fig. 1 and Fig. 2 describe the general features of the colon. Fig. 3 provides information on mucin staining, which offers nothing new and just confirms previously published data.

Fig. 4 provides information on the differential expression pattern of certain genes, but the data relate to wild type mice and the double-mutant mice. This provides no clue as to whether the observed difference in the expression of any gene is due to Muc2 mutation or ApcMin mutation.

Critique

  1. Supplemental Fig. 1 shows that there are 10 genes that are unregulated in Winnie mice compared to WT. There are also 10 genes that are up regulated in Winnie/ApcMin mice compared to Winnie mice. Are these 10 genes the same between the two groups?
  2. Supplemental Fig. 1 shows 10 genes that are differentially unregulated in Winnie/ApcMin mice compared to Winnie mice, but the data in Fig. 4 show 18 genes that are upregulated. Aren't the data in the Suppl. Fig. 1 from the same qPCR experiment?
  3. The same concern arises for the differentially expressed genes between ApcMin mice and Winnie/ApcMin mice. 
  4. The qPCR experiment has been done with RNA isolated from whole colon. It would have been better if the experiment had been done with RNA isolated from epithelial cells rather than the whole colon.
  5. Is there any evidence of disruption of intestinal/colonic permeability in double-mutant mice compared to WT mice? This has been documented in previously published reports for Winnie mice. How about in the double-mutant mice? Does it change?
  6. Why did the study get terminated at 5 weeks of age for the mice? What happens if the mice are allowed for a few more weeks? Could the polyps in the distal colon become easily visible for a more detailed analysis?
  7. As it stands, the present manuscript focuses on a potentially interesting and important project, but the study has been conducted with a narrow scope. A more detailed analysis of the mouse model is needed.

Author Response

We thank the reviewer for the constructive comments.

We revised the manuscript to address reviewers concerns.

We mainly worked to better explain the molecular data (Line 151-165) and report ACF progression from 5 to 8 weeks “2.4. Winnie-ApcMin/+ model confirms the multi-step nature of CRC increasing the multiplicity and tumor grading of dysplastic ACFs in older 8-wk-old mice.” (Line 169-198; Figure 5 and Table 2)

Unfortunately, we can’t address the reviewer question 5 as our animal protocol does not allow testing intestinal permeability. We submitted a new request for animal study, but we suspect that the eventual approval will be postponed due to the COVID-19 emergency.

We believe that thanks to the reviewers comments the manuscript has been significantly improved and we now hope that could be considered suitable for publication.

Reviewer 2 Report

In the manuscript “Winnie-APCMin/+ mice: a spontaneous model of colitis-associated colorectal cancer combining genetics and inflammation” by de Santis et al., the authors crossed two mouse strains, the APCMin/+ and the Winnie, with each other in order to create a very aggressive colon cancer mouse model with a very high penetration rate.

The experiments are straight forward and well performed.

The new mouse model is attractive due to its very high penetration rate of colon cancer and the severity of disease. The model is physiological relevant and has the clear potential to become a gold standard model for colon cancer research. The only limitation of this novel model appears to be the extremely early onset of colon cancer incidences. Such a model limits the opportunities for treatment. Therefore, an inducible induction of Winnie mutation would be very attractive.

Minor suggestion for improvement: Please introduce the underlying mutation for Winnie in the Abstract.

Author Response

We thank the reviewer for the positive and constructive revision.

In the present revised version, we improved the manuscript following reviewers’ suggestion and hope that he will now consider it suitable for publication.

Round 2

Reviewer 1 Report

I appreciate the effort by the authors to improve the manuscript by adding data for another time point. I also see their point as to why they cannot perform the experiments related to intestinal permeability. 

I agree that the manuscript has been certainly improved because of the additional new data.